# Tissue Resident and Infiltrating Immune Cells: Their Influence on the Demise of Beta Cells in Type 1 Diabetes

**DOI:** 10.3390/biom15030441

**Published:** 2025-03-19

**Authors:** Sophie L. Walker, Pia Leete, Joanne Boldison

**Affiliations:** Department of Clinical and Biomedical Sciences, University of Exeter, RILD Building (Level 4), Barrack Road, Exeter EX2 5DW, UK; s.walker8@exeter.ac.uk (S.L.W.); p.leete@exeter.ac.uk (P.L.)

**Keywords:** type 1 diabetes, pancreas, beta cells, immune cells, communication

## Abstract

Type 1 diabetes (T1D) is an organ-specific autoimmune disease that results in the selective loss of pancreatic beta cells and an eventual deficit in insulin production to maintain glucose homeostasis. It is now increasingly accepted that this dynamic disease process is multifactorial; involves a variety of immune cells which contribute to an inflamed pancreatic microenvironment; and that the condition is heterogenous, resulting in variable rates of subsequent beta cell damage. In this review, we will explore the current understanding of the cellular interactions between both resident and infiltrating immune cells within the pancreatic environment, highlighting key mechanisms which may promote the beta cell destruction and islet damage associated with T1D.

## 1. Introduction

Type 1 diabetes (T1D) is an autoimmune disease which develops because of contributing genetic and environmental factors and leads to a defect in self-tolerance, resulting in the T-cell mediated destruction of pancreatic beta cells and a subsequent loss of insulin production. T1D is now stratified into distinct aetiological stages, with the presence of multiple islet autoantibodies considered as Stage 1, providing a robust readout for the induction of beta cell autoimmunity [1]. Stages 1 and 2 are associated with immune infiltration of the pancreas, and there is a large body of evidence that autoreactive CD8^+^ T-cells are directly involved in the killing of beta cells and the progressive destruction of islets [2,3].

In this review, we will explore aspects of our current understanding surrounding the interactions between these site-specific immune cells and the endocrine compartment. Specifically, we will consider the interplay between resident leukocytes, beta cells and the heterogeneous immune cell infiltrate observed to influence the rate and severity of beta cell demise. We will consider immune dynamics during disease progression at the islet level, the study of which is providing ever more granular insight into the etiopathology of T1D in the pancreas. Important mouse studies are also included, but our focus is primarily on studies of the human pancreas, through which we are now able to offer greater insights about immune cell dynamics in human disease via the emergence of novel technologies such as spatial transcriptomics.

## 2. The Immune Landscape of the Exocrine Pancreas

Individuals living with T1D exhibit reduced exocrine function and their pancreas is notably smaller than healthy individuals: the difference in mass is too pronounced to be accounted for only by the loss of beta cells which, at best, likely comprise ~0.65–0.85% of normal pancreatic mass [4,5]. In T1D, the exocrine tissue also contains significantly large populations of immune cells, including myeloid cells and CD8^+^ T-cells [6,7,8]. However, tissue-specific CD8^+^ resident-memory T-cells (T_RM_) have also been identified in the healthy human pancreas [9,10]. These differ from the circulating CD8^+^ T-cell memory pool by the unique expression of a tissue-residency transcriptional signature, including CD103 (αE integrin) and CD69. The T_RM_ isolated from the pancreas in healthy adult donors [9,11] can also express PD-1 [11], and the cellular interactions between T_RM_ and localised macrophages, via the immune checkpoint PD-1/PD-L1 pathway can impact their functional regulation [11], depicted in Figure 1A.

Resident macrophages populate most organs and act as sentinels, surveying for foreign pathogens and playing a role in tissue homeostasis [12], and are further discussed in Section 3. Notably, the proportion of resident macrophages is significantly smaller in the human pancreas compared to that observed in mice [11,13,14]. Mouse studies suggest that stromal macrophages, situated in the parenchyma, are more akin to an M2-like (regulatory) phenotype [14]. Blockade of the PD-1 pathway in human CD8^+^ T_RM_ and macrophage co-cultures resulted in the increased capacity to produce IFN-γ, TNF-α and IL-2 [11]. Thus, disruption of PD-1/PD-L1 signaling, which is observed in individuals with T1D [15] may shift T_RM_ cells from regulators to a more effector phenotype. Upon an antigen encounter, the T_RM_ cells can become particularly effective at producing cytotoxic granules and inflammatory cytokines, facilitating rapid clearance of infections [16]. Stimulation of T_RM_ cells isolated from a healthy human pancreas also show enhanced IFN-γ and IL-2 production, compared to circulating effector memory CD8^+^ T-cells [11]. Therefore, contrary to their role in health surveillance, in dysregulated environments T_RM_ cells may play a role in exacerbating T1D when they encounter autoantigens, as demonstrated in psoriasis [17]. Many of the T_RM_ in the healthy human pancreata are specific for preproinsulin (PPI), a beta cell antigen [10], and thus we could speculate may result in misdirected-targeted beta cell destruction when under inflammatory stress.

Another cell of note is the ductal cell, which has a specific role in immune cell recruitment via the secretion of pro-inflammatory cytokines [18]. A highly significant enrichment of the gene signature DC1 was identified in the ductal cells of T1D donors; as these cells lack expression of co-stimulatory factors CD80 and CD86, this may suggest an attempted decoy role for DC1 cells (to deactivate CD4^+^ T-cells) in T1D [19]. Supporting this notion, the authors observed that HLA-DR expressing ductal cells were surrounded by CD4^+^ T-cells and CD11b^+^ myeloid cells [19]. Now, with novel multiplex-techniques to study the heterogeneity of pancreatic resident cells in both the steady state and in the development of T1D, we can further dissect these dynamics.

## 3. The Role of Intra-Islet Macrophages in the Context of T1D

Mouse studies have shown that in the steady state, macrophages within the pancreatic islets express MHC II and are more M1-like in profile (more inflammatory), than their stromal counterparts [14]. Islet macrophages can, therefore, process insulin transferred from beta cells and present to CD4^+^ T-cells [20]. Furthermore, intra-islet macrophages can respond to signals from beta cells, specifically sensing ATP through purinergic receptors [13], thus providing an important route for metabolic and cellular monitoring [13,21]. Indeed, spatial studies in mice show phagocytes in close contact with beta cells and with blood vessels [20], where they also sense blood borne stimuli [22].

Early depletion of resident macrophages in mice delays the onset of autoimmune diabetes, and a reduction of infiltrating leukocytes demonstrates an important role for resident macrophages during initiation of the disease [23]. However, what remains unclear is how these resident antigen presenting cells (APCs) are influenced by the inflammatory environment present during disease progression. It is possible these resident APCs undergo differentiation and exacerbate beta cell destruction via mechanisms usually associated with circulating tissue infiltrating macrophages (see Section 6.2). Human resident islet macrophages secrete IL-1 in response to proinflammatory stimuli such as IFN-γ [24] (Figure 1B), which could contribute to ongoing beta cell damage observed under stress, including viral infection. In NOD mice, islet resident macrophages undergo phenotypic changes as mice age and develop diabetes, expressing less CD39 which modulates their regulatory phenotype [25]. Furthermore, islet macrophages have a basal activation signature, irrespective of autoimmune susceptibility [26]. A key part of this signature is expression of CXCL16, where NOD mice lacking Cxcl16 are protected from autoimmune diabetes via a mechanism that impairs oxidised low-density lipoprotein (OxLDL) clearance, subsequently disrupting the effector function of pathogenic CD8^+^ T-cells [26]. Taken together, this suggests that intrinsic expression of a proinflammatory profile in islet resident macrophages, under certain circumstances, can permit autoimmunity.

## 4. Intrinsic Beta Cell Mechanisms and Visibility

It is now well established that the stress, dysfunction and intrinsic heterogeneity of beta cells influences their own destruction, and this subject is extensively reviewed elsewhere [27,28,29]. Here, and in Figure 1B, we highlight the principal mechanisms by which beta cells may become visible to autoimmune attack. The hyperexpression of HLA class-I is now accepted as one of the “hallmarks” of T1D [30,31]. Interestingly, in islets with residual insulin, all endocrine cells display this elevated expression, and it is presumed this is driven by the upregulation of interferon-related signalling pathways, which may drive elevated HLA in either a paracrine or autocrine manner, and yet only the beta cells are targeted. It has been postulated that other endocrine cells are more resilient to immunological killing by virtue of having different anti-apoptotic machinery. For example, anti-apoptotic BCL2L1 is more abundant in alpha cells, versus higher levels of pro-apoptotic CHOP being expressed in beta cells [32], and similar differences may exist across the other islet cell types. Moreover, in humans, HLA class-I upregulation precedes the influx of invading immune cells, and diminishes upon beta cell ablation from a given islet (reviewed previously [33]). HLA expression can occur as the result of a stressed cell responding to an inflammatory environment or insult, such as a viral trigger. Intriguingly, and possibly critically, it was demonstrated that any evidence of viral infection in islets is restricted to beta cells [34].

Other intrinsic mechanisms, such as expression of TET2 is increased in beta cells from donors with recent onset T1D and autoantibody positive individuals prior to T1D onset which may deliver signals to immune cells to direct targeted killing [35]. Furthermore, *TET2* can be induced by cytokine production (IL-1β or IFN-γ) by inflammatory infiltrates [35]. Another beta cell stress response associated with inflammation is the secretion of CXCL10 [36]. This chemokine can induce apoptosis via the TLR4 receptor [37], and also act as a chemoattractant for CXCR3^+^ T-cells, albeit to varying degrees depending on the individual [38]. CCL2, also released by stressed beta cells was increased in isolated human islets after IL-1 exposure [39,40]. It was also shown to promote the recruitment of monocytes in transgenic mice, and higher levels of CCL2 were associated with increased diabetes incidence, independent of T- and B-cell presence in the islets [41]. However, in transgenic mouse models, this CCL2-mediated recruitment of macrophages, when in the presence of CD11c^+^CD11b^+^ DCs, has been shown to dampen inflammatory responses, via DC-directed diabetogenic T-cell inhibition [42]. In line with this, the PD-L1/PD-1 axis acts to defend the islets against T-cell immune infiltration [43] and is upregulated by the release of factors such as IFN-γ and -α from immune cells [15] in a counter regulatory mechanism from beta cells.

## 5. The Insulitis Trajectory and Immune Networks

It is important to consider that beta cells, which are situated in the pancreatic islets of Langerhans, are physically separated from the surrounding parenchyma by a peri-islet basement membrane and an interstitial membrane forming the peri-islet capsule, creating over two million discrete ‘immune privileged’ sites in a single pancreas [44]. Therefore, in addition to the relative inaccessibility, studying immune:target-cell interactions is a challenge due to the rapid immune synapse dynamics *in situ*. However, studies are now using live tissue slice technologies, and advanced image capture and analysis techniques are being developed [45]. Recently, imaging mass cytometry (IMC) and neighbourhood analysis supported that the interactions between CD8^+^ T-cells and beta cells at the islet level are rare in tissue from T1D donors [46]. Although, when interactions between T-cells and beta cells were observed they were most prominent in inflamed islets still containing beta cells, and in recently diagnosed individuals [46].

We can now exploit these powerful novel technologies and apply them to pancreatic tissues collected via multiple biobanks and organ donation programs, which recently culminated in several studies assembling a pseudotime trajectory to investigate immune interactions at the local level [46,47,48] (see Figure 2). Damond et al. (2019) [46] studied frequent immune cell associations based on spatial locations in pseudostage 2 and 3, as islets progress towards complete beta-cell ablation [46]. At pseudostage 2 (islets with remaining functional, but altered, beta cells), most of the cell associations are between cytotoxic T-cells and either naïve cytotoxic T-cells, monocytes or ‘other’ immune cells; in contrast, in pseudostage 3 (islets devoid of beta cells), the highly associated cells are neutrophils and monocytes, and cytotoxic T-cells and T helper cells/‘other’ immune cells [46]. Similarly, Barlow et al. (2024) used a neural network to quantify islet substates, whereby the progression of inflamed islets was associated with CD8^+^ T-cell functionality [48]. Analysis of cellular neighbourhoods revealed three distinct populations: neighbourhood 1: (CD8^+^T-cells|B-cells); neighbourhood 2: (Macrophage|Stromal Cells|B-cells); and neighbourhood 3: (Vasculature|B-cells). Neighbourhood 1 was often next to 2 and 3; however, neighbourhoods 2 and 3 were rarely adjacent to each other [48]. The CD8|B cell neighbourhoods are suggested to be immature TLSs in the tissue (discussed in Section 6.4). The latter communication network between B-cells and T-cells in the pancreas is well accepted in both humans [48] and in mice [49,50] and is further evidenced by the observation that insulitic CD8^+^ T-cells and B-cells are in proximity in regions with residual beta cells, in pancreas tissue from donors with T1D [47,51]. Moreover, the expression of the proliferation marker Ki67 in both the B-cells and CD8^+^ T-cells is markedly increased, compared to donor tissues without diabetes [47]. In Figure 2 we illustrate some of these observations in the context of disease progression at the islet level.

## 6. Components of Pancreatic Insulitis in T1D

Fully characterising the relevance of each immune element in the development of T1D is hampered by the challenge of acquiring peripheral blood samples and biopsies from the same individual, largely due to the difficulty in collecting pancreatic biopsies [52]. This is a significant barrier when aiming to correlate peripheral blood and tissue-based findings to inform our understanding of the different immune population dynamics in the pancreas in T1D during the progressive disease stages, and across different demographics. In this part of the review, we will focus on the immune cells well documented in human insulitis and their role in beta cell destruction, with key insights summarised in Table 1. Cells that may be integral to protective mechanisms in the pancreas and are few in numbers such as FoxP3^+^ Tregs [7] have not been detailed here but should be noted.

### 6.1. Neutrophils

The role of neutrophils in T1D has, compared to other immune cells, been less studied but recent reviews have highlighted their relevance [65,66,67,68]. In peripheral blood, neutrophil counts are lower in individuals positive for islet autoantibodies [52,69] and those recently diagnosed [52,70], compared to healthy controls; this additionally correlated to reduced beta cell function [52]. The significant reduction in circulating neutrophils corresponds to an observed infiltration into pancreatic tissue, predominantly within the first year of diagnosis in T1D patients [54]. Neutrophils, unlike subsets of lymphocytes, were reported to remain elevated in the tissue of individuals with long-duration T1D [46], suggesting that in some cases the persistence of neutrophils remains after onset.

In-keeping with their role as the immune system’s “first responders”, pancreas-residing neutrophils were identified in presymptomatic autoantibody positive individuals (Stage 1) [53], suggesting that they may be early sentinels of disease induction. Neutrophils are also detected early in the islets of NOD mice, with peak accumulation being observed at 8-weeks of age and then a decline in neutrophil numbers thereafter [53]. Furthermore, neutrophils are involved in the activation of plasmacytoid dendritic cells (pDCs) and subsequent initiation of autoimmune diabetes [71]. In contradiction to this hypothesis, gene analysis showed clustering of a neutrophil gene, *CXCR1*, in slow-progressors, and an increase in neutrophil-associated gene expression with age [54]. Curiously, *CXCR1* inversely correlates with the B-cell gene *CD19* [54], and increased B-cell signatures are associated with a rapid, young-onset of T1D [51].

Importantly, the pancreatic neutrophils do not reside in clusters around or near to the beta cells, but are uniformly distributed throughout the pancreas and within vessels in both T1D donors [72] and in presymptomatic autoantibody positive donors [52]. Investigation into two individuals, *post-mortem,* showed neutrophils localised at a greater density around pancreatic ducts and throughout pancreatic parenchyma following initial symptoms, whilst after 3–4 weeks, neutrophil counts were similar to controls [73]. The presence of neutrophil extracellular traps (NETs) in the tissue suggests neutrophils can contribute to beta cell death via NET formation which can damage tissue, inducing beta cell stress, and promote inflammation (see Figure 2) [52,67,74].

### 6.2. Macrophages and Dendritic Cells (DCs)

Macrophages and DCs provide a missing link between distressed beta cells and the activation of the adaptive immune system, including T- and B-cell involvement [21,71]. Research using the NOD mouse model demonstrates the importance of both macrophages and DCs in antigen presentation, lymphocyte recruitment and the development of autoimmune diabetes [71,75,76,77]. Macrophages can become highly activated when responding to IFN-γ from activated T-cells and can attack the beta cells via nitric oxide and IL-1 release [55]. Recently, in NOD mice it was demonstrated that fluctuating waves of DCs reside in the pancreas during the initial pre-symptomatic development and onset of autoimmune diabetes [53]. This influx of pancreatic DCs was preceded by an increase in DC frequency in the pancreatic lymph nodes (PLN), whilst the number of circulating DCs remained constant, suggesting that there is dynamic trafficking of DCs between the two organs during disease development [53,78].

Human studies have also identified an altered presence of DCs in the peripheral blood of people with T1D [79], and have been proposed as a potential biomarker for identifying patients who may have a partial remittance/honeymoon phase soon after diagnosis [80]. Both DCs and macrophages are observed in the inflamed pancreatic tissue [7,47,81]. Macrophages have been detected in early infiltrates [7], in islets with or without beta cells [47], and appear not to correlate with inflamed islet clusters ([48]; peer-reviewed preprint, peer-review reports available online). Furthermore, macrophages produce both IL-1β and TNF-α regardless of the presence of remaining beta cells, both of which can have detrimental effects [81]. However, DCs in the human pancreas tissue are relatively understudied [21]. Single cell studies in NOD mice reveal macrophages and DCs are influenced by the pancreatic inflamed microenvironment and change within the tissue alongside the progression of autoimmunity [64]. Understanding how autoimmune progression imprints on these cellular subsets will help define their importance in the tissue at the different stages of diabetes progression. Particularly, the differential expression of TLRs that sense specific pathogens and the subsequent downstream signaling inflammatory cascades will be important, as TLR3 expression is observed on macrophages in pancreatic tissue of deceased donors diagnosed with T1D [82].

### 6.3. Natural Killer (NK) Cells

Although research into the relevance of NK cells in T1D has been undertaken in peripheral blood samples and a decrease of NK cells is present in donors with T1D [83], the presence of NK cells in the pancreata of T1D individuals has been less widely studied and their role in T1D is contested [84,85]. Studies quantifying the levels of NK cells in pancreatic infiltrates have shown that pancreata from T1D donors [7] have similar distribution and presence to that seen in a healthy pancreas [9]. Although, some pancreatic studies have also noted an increased presence in NK cells present in pancreatic tissue donors with a diagnosis of under 2 years, albeit in very small numbers [47]. Further studies with longitudinal analyses of the presence of NK cells in pancreatic tissues will be needed to further elucidate whether the small population of NK cells have a currently unknown pivotal role in T1D development or progression.

### 6.4. B Lymphocytes

There has been renewed interest in the role of B-cells in T1D following a seminal study by Leete et al. in 2016 [51] which identified heterogeneity in the pancreata of T1D donors and further work has since developed this understanding of differing disease progressions [86]. A relationship between B-cell infiltration and the age at onset and severity of beta cell loss is observed [51], and recently reviews have highlighted current studies in B-cells in T1D [87,88].

So far, the characterisation of B-cells within the human pancreas is limited. One recent highlight is the observation that insulin-reactive B-cell subsets (B_ND_2 cell phenotype) are enriched in the PLN of young-onset donors with T1D [89]; however, no analysis of the pancreatic tissue to determine the presence of B_ND_2 was undertaken. NOD mouse studies highlight that B-cell subsets residing in the pancreas are transcriptionally different to B-cells situated in the PLN at the age of disease onset [90]. Investigation into the phenotype of B-cells is currently one of the most important questions to address; we are yet to know whether B-cells in the pancreas are of a specific subset or if they phenotypically or functionally differ depending on the age of disease onset. Use of current cutting-edge technologies such as single cell spatial transcriptomics and high-throughput multiplex immunohistochemistry will help reveal whether B-cells adopt a different expression profile in the pancreas and uncover the importance of these cells in the tissue.

Curiously, plasma cells (using CD138 as a marker) are infrequent in the pancreatic infiltrate at all stages of insulitis [7], but were identified in tissues containing tertiary lymphoid structures (TLSs) [62], sites known for antibody production and class-switch recombination (see Figure 2) [91]. Still, considering autoantibodies are a key biomarker for islet autoimmunity, with autoantibodies appearing before clinical onset, we are yet to detect isotype-specific B-cells in the tissue. Future investigation on the specificity of B-cells is required, noting that in childhood, insulin-specific islet autoantibodies (IAAs) are more prevalent, but not limited to IAA [92]. A younger age of onset was associated with the presence of TLSs, and correlated with increased insulitis; however, only four donors showed compartmentalised B- and T-cell zones, signifying a more mature TLS [62]. However, immature TLSs were identified both adjacent and non-adjacent to islets in a recent study by using CO-Detection by indEXing (CODEX) tissue imaging and cellular neighbourhood analysis [48]. CD45RA^+^ CD8^+^ T-cells were enriched in the immature TLS, compared to the rest of the pancreatic tissue, and these T-cells are thought to traffic to the TLS through high endothelial venules (HEVs) [48]. The presence of TLSs is hypothesised to contribute to the prolonged presence of inflammation in human T1D [62], but understanding more about TLSs may provide more insight into tissue specific B-cell functionality and local autoantibody production.

### 6.5. T Lymphocytes

Numerous studies identify T-cells as the major component of pancreatic insulitis and are present in islets at a significantly higher density in individuals affected by T1D or autoantibody positivity, particularly increasing with disease progression [7,93]. In human pancreata studies, CD8^+^ T-cells are predominant, with CD4^+^ T-cells fewer in number, but both populations expand as beta cell loss from islets progresses [51]. Autoreactive CD8^+^ and CD4^+^ T-cells, specific to beta cell antigens, are observed in the pancreata of donors diagnosed with T1D [10,94,95,96]. At onset, the localised autoreactive T-cells in inflamed islets are more likely to exhibit a single specificity, but diversifies with increasing disease duration [94]. CD4^+^ T-cells that are reactive to proinsulin have been extracted from inflamed human pancreatic islets [95] and CD8^+^ T-cells specific for pre-proinsulin (PPI) [10,94] are largely located at ICIs, as opposed to insulin deficient islets (IDIs) [10]. Taken together, alongside the observation that CD8^+^ T-cells are found in the pancreas regardless of the age of diagnosis, and are always the most populated immune cell in the insulitic lesion, CD8^+^ T-cells are most likely to be the main cytotoxic mediator [51]. However, the heterogeneity of T-cell reactivity in individuals during different stages of disease, or with different ages of onset is yet to be determined.

A recent paper focuses on some of the differences between infiltrated islets and their non-infiltrated counterparts, and the surrounding extra-islet tissue [48]. Interestingly, some IDIs still had CD8^+^ T-cells present within their boundary, suggesting that the specific attractants for islet CD8^+^ T-cells are still present post-beta cell death [48]. This persistence after beta cell death goes against current thinking and is evidence that CD8^+^ T-cells are only present around insulin^+^ islets [94]; this difference could be explained by the omission, in the study by Barlow et al. (2024) [48], of alternative beta cell markers (i.e., *Pdx1* and *Nkx6.1* co-expression) to identify insulin-deficient beta cells still present in the tissue, retaining lingering T-cells.

The T-cells within islets in individuals with T1D exhibit heterogeneous activation profiles and comprise of both a memory and naïve phenotype [9,46]. In NOD mice as early as 4-weeks of age, CD4^+^ T-cells show a degree of heterogeneity, with both regulator and effector cells present in the pancreas, while CD8^+^ T-cells show functional-related changes indicating a response to the local environment [64]. In line with this, substates are seen throughout different islets within the same individual, with the composition of CD8^+^ T-cells being attributable to the islet microenvironment, not the extra-islet space surrounding the islet [48]. In biopsy specimens from adult individuals with T1D, 43% of CD8^+^ T-cells surrounding insulitic islets display a resident phenotype (CD69^+^CD103^+^), characterised by a lack of cytotoxic-associated genes and a more cytokine-driven immune response [97]. Further work is required to fully understand if T-cell phenotypes are altered during disease progression in the human pancreas, and whether the heterogeneity of the disease is reflected in the T-cell functionality, which will be an important step in further dissecting how to stratify therapeutics.

## 7. Conclusions

Altogether, understanding more of how the beta-immune cell dialogue is disturbed in diabetes development, in different individuals, is crucial to developing better therapeutics to halt disease progression. Combining observations of how beta cells create an escalating loop of their own demise with the different progressive disease stages, at the islet level, will help stratify the time at which to intervene with a specific treatment. Detailing beta cell level trajectories, shown in Figure 2, may help elucidate the key islet:immune mechanisms not yet appreciated. This is of particular importance considering recently evolving observations that functional defects in beta cells are independent of T-cell infiltration at the islet site, as defects in the beta cells are observed in both infiltrated and non-infiltrated islets in human pancreata with T1D ([98]; preprint). Future efforts to investigate such observations will help clarify which mechanisms induce such defects in beta cells, and how the immune system responds to such changes.

Moving forward (and especially with growing acceptance that T1D is impacted by a variety of demographic features including age and ancestry), it will be key to fully characterise these functional interactions in the context of individual demographics and across various populations. It will also be imperative to consider the immune, exocrine and endocrine T1D-associated risk genes linked with specific cell types that may help interpret cellular changes at the single cell level within the islet [99]. Considering novel analysis techniques, predictive modelling and artificial intelligence is now providing ever more refined opportunities to study and predict disease-associated features in the human pancreas. These tools will allow us to examine combinations of immune cell subsets, with distinct phenotypes in the same individual and address why some beta cells are targeted over others.

Solutions will, indeed, need to be found to address the current limitations of imaging resolution, including live tissue imaging in humans, and the ability to navigate complex datasets; however, given the current pace of progress, the field is poised for significant advancement. New therapies directed at halting the progression of disease, and now the use of anti-CD3 treatment, will allow us to further stratify and understand the differential responses to therapy, helping to bridge the gaps in knowledge around when and who to treat, and with which intervention.

## Figures and Tables

**Figure 1 biomolecules-15-00441-f001:**
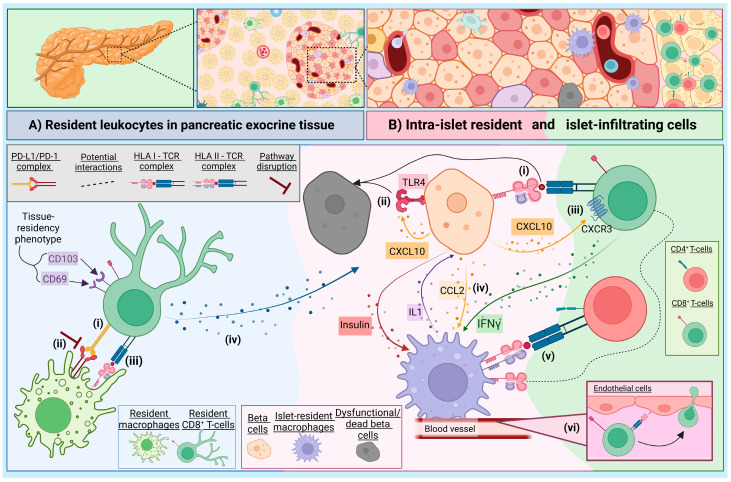
Resident cell-to-cell interactions in the exocrine tissue and islet compartments, in the context of type 1 diabetes (T1D). (**A**). (**i**) In healthy tissue, CD8^+^ memory T-cells (CD69^+^CD103^+^) patrol and survey for foreign antigens and are regulated by M2-like myeloid cells via the PD-1/PD-L1 axis. In T1D, these macrophages may contribute to the inflammatory environment (**ii**) if the PD-L1/PD-1 complex formation is disrupted, or (**iii**) by presenting beta cell antigens to islet-specific CD8^+^ T-cells via HLA-I: T-cell receptor (TCR) complexes, both of which could result in the subsequent (**iv**) release of effector cytokines, such as IL2, IFNγ and TNF-α, by CD8^+^ T-cells and the misdirected targeting of beta cells by resident CD8^+^ T-cells. (**B**) Intra-islet resident cells, myeloid cells and beta cells themselves, participate in beta cell death. (**i**) Beta cells hyper-express HLA class-I molecules, increasing their visibility to CD8^+^ T-cells. Stressed beta cells release CXCL10, which interacts with (**ii**) TLR4 and is detrimental to beta cell viability, and (**iii**) attracts CD8^+^ T-cells to the beta cells via the CXCL10:CXCR3 axis. (**iv**) The release of CCL2/MCP-1 from beta cells and IFN-γ from activated T-cells results in macrophages releasing IL-1, further damaging beta cells. (**v**) Islet-resident macrophages express HLA class-II and have a more M1-like inflammatory profile; they process insulin transferred from beta cells and present this to CD4^+^ T-cells to elicit an immune response. (**vi**) CD8^+^ T-cells undergo transendothelial migration to gain access to the pancreas from general circulation. Created in BioRender. Walker, S. (2025) https://BioRender.com/b50t774 (accessed on 16 February 2025).

**Figure 2 biomolecules-15-00441-f002:**
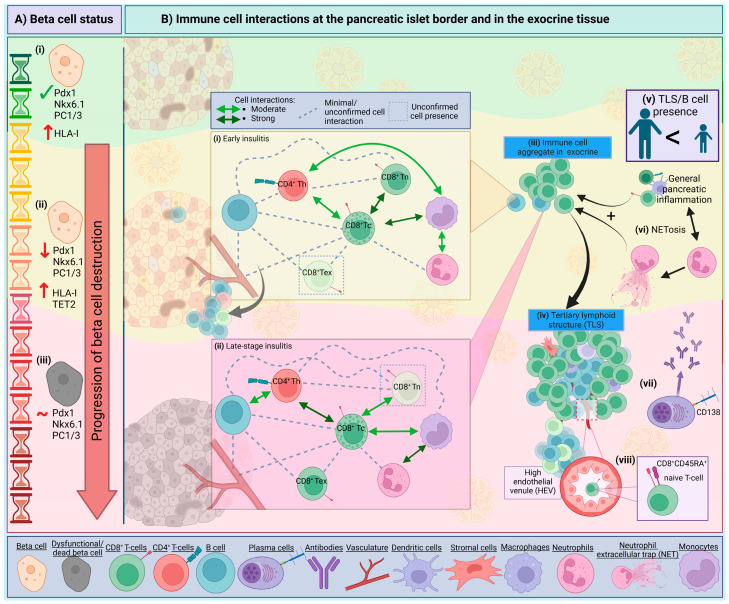
Immune cell networks and beta cell alterations in the trajectory of islet destruction. Throughout T1D development, pancreatic islets undergo substantial changes, signified by a progressive beta cell destruction coinciding with the presence of immune cell aggregates. The trajectory of beta cell changes is depicted in (**A**). (**i**) An abundance of beta cell markers and a high mass within islets, but increasing HLA-I expression (green, top panel) to (**ii**) having a reduction in beta cell markers, and elevated HLA-I and TET2 expression (yellow, middle panel) to (**iii**) becoming dysfunctional and dying, having a reduction in beta cell mass within islets but with a stable level of beta cell markers (pink, bottom panel). Corresponding with these beta cell changes is the appearance of immune cells at the islet border (**B**) which create a network of interactions which are primarily associated with islets with remaining beta cells. (**i**) Strong interactions between cytotoxic CD8^+^CD45RO^+^ T-cells (CD8^+^ Tc) with naïve CD8^+^CD45RA^+^ T-cells (CD8^+^ Tn) have been observed, while only moderate interactions are identified between CD8^+^ Tc with CD4^+^ T-helper cells (CD4^+^ Th), CD4^+^ Th with monocytes, and monocytes with neutrophils. CD8^+^ Tc and monocytes are also strongly associated. As beta cell destruction progresses (**ii**) islets are more insulin deficient, and the immune networks are altered. At this stage, there may be a diminished presence of CD8^+^ Tn with a lowered interaction with CD8^+^ Tc, and an increase in CD8^+^ Tex (exhausted T cells); CD8^+^ Tc strongly interact with CD4^+^ Th, but their interactions with monocytes is now reduced. Monocytes are now more likely to interact with CD4^+^ Th cells and neutrophils. All other interactions not stated here, in either stage of beta cell destruction, are either minimal or not described. (**iii**) Lymphoid aggregates and (**iv**) tertiary lymphoid structures (TLSs) at the islet boundary and the exocrine tissue may be similar in cellular composition; (**v**) TLSs and high levels of B-cells are seen primarily in individuals with a younger onset of disease, however not exclusively. (**vi**) Neutrophils play a role via the development of neutrophil extracellular traps (NET) and consequent NETosis, and interacting with T-cells, both of which cause extensive T-cell expansion. (**vii**) CD138^+^ plasma cells are observed in TLSs and could be a source of local antibody production. (**viii**) CD45RA^+^ naïve CD8^+^ T-cells have been identified in TLSs and are thought to traffic from high endothelial venules (HEVs), a characteristic of TLSs, and are hypothesised to then traffic to the pancreatic islet border. Dashed lines and boxes represent minimal/unconfirmed interactions and cell presence, respectively. Double-ended arrows represent interactions between cells: dark green shows strong interaction, and light green shows a moderate interaction between cells. Created in BioRender. Walker, S. (2025) https://BioRender.com/i43w255 (accessed on 16 February 2025).

**Table 1 biomolecules-15-00441-t001:** Key insights regarding cellular components of pancreatic insulitis.

Cell Type	Human or Mouse	Study Highlight	Suggested Role in Beta Cell Destruction/Targeting	References
Neutrophils	NOD mouse	Peak accumulation of neutrophils is at 8-weeks old, then a progressive decline is observed	Interact with CD8^+^ T-cells and monocytes at different disease stages resulting in enhanced T-cell responses and expansion	[53]
	Human	Reduction in peripheral, circulating neutrophils correlates with the infiltration of neutrophils into the pancreatic tissues, particularly in the first year of diagnosis		[54]
Macrophages	NOD mouse	In response to IFN-γ from activated T-cells, macrophages become highly activated and attack beta cells via IL-1 and nitric oxide release	Release cytokines to effect beta cell health and interact with CD4^+^ T-cells via HLA-II TCR complexes	[55]
	Human	Detected in early infiltrates, associated with both ICIs and IDIs, and not correlating with inflammatory clusters		[47,48]
Dendritic cells (DCs)	NOD mouse	Influx of pancreatic DCs is preceded by and increase in DC frequency in the pancreatic lymph node (PLN)	Presentation of islet antigens to result in further immune cell attack of beta cells, typically by CD8^+^ T-cells	[53]
	Human	Relatively understudied in the human pancreas		[21]
Natural killer (NK) cells	Mouse	NK cell depletion was shown to significantly decrease the incidence of T1D	Due to the small sizes of the cell populations currently, the effect and interactions of other cells with NK cells is relatively unknown	[56]
	Human	Increased presence in NK cells in pancreatic tissue donors with a diagnosis of under 2 years, although the populations are very small		[47]
Innate lymphoid cells (ILCs)	Mouse	Reduction in IL2-producing gut ILC3s and Tregs is associated with insulitis presence in the pancreatic tissues; IL-22 production induces islet-associated β defensin mBD14 which can protect NOD mice from autoimmune diabetes.	Modulate cytokine production in the gut to affect beta cell survival in the pancreas	[57,58]
	Human	Data from T1D pancreatic tissues is limited		[59]
B lymphocytes	TLR7-knockout, and NOD mice	TLR7 knockout mice have a delayed onset and reduced incidence of type 1 diabetes development, with reduced B-cell function; TLR7 gene and protein expression is enriched in B-cell subsets	Present antigens to attract T-cells to increase beta cell destruction	[60,61]
	Human	TLSs are associated with young age of onset; compartmentalised B- and T-cell zones were only present in a limited number of recent onset donors but also observed in a long-standing donor in an individual case-study		[62,63]
T lymphocytes	NOD mouse	CD4^+^ effector and regulatory T-cells present as early as 4-weeks of age, while CD8^+^ T-cells respond to local environment via functional-related changes	Direct cell-surface receptor interaction with beta cells to elicit beta cell destruction	[64]
	Human	Substates of T-cells and the variable composition of CD8^+^ T-cells between islets in the same individual have been identified, in response to islet microenvironment		[48]

## Data Availability

Not applicable.

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
