# Peer review of "Tissue Resident and Infiltrating Immune Cells: Their Influence on the Demise of Beta Cells in Type 1 Diabetes"

_biomolecules, 2025, doi:10.3390/biom15030441_

Round 1

Reviewer 1 Report

Comments and Suggestions for Authors

In this review titled” Tissue Resident and Infiltrating Immune Cells: Their Influence on the Demise of Beta Cells in Type 1 Diabetes” the authors have comprehensively summarized the immune cell interaction with healthy and diseased beta cells and their role in exacerbating inflammation leading to diabetes progression. In many places, the authors have theorized the mechanism lacking much evidence to support it. However, these can be areas for future research and must be carefully considered by the readers. The authors have justly discussed the various players involved in the progression of diabetes, however the depth was missing. The artwork to demonstrate the interaction of beta cells and immune cells was very informative and summarized the body of work effectively. One key feature that was lacking was the role of natural killer cells which also has been shown widely as vital players in diabetes development. Another missing link is how innate lymphoid cells (ILCs) play a role in the pathogenesis of type 1 diabetes. The authors have not discussed or even acknowledged these limitations.

Comments on the Quality of English Language

The English could be improved to more clearly express the research.

Author Response

Responses are in blue text, under each numbered comment.

In this review titled” Tissue Resident and Infiltrating Immune Cells: Their Influence on the Demise of Beta Cells in Type 1 Diabetes” the authors have comprehensively summarized the immune cell interaction with healthy and diseased beta cells and their role in exacerbating inflammation leading to diabetes progression.

  1. In many places, the authors have theorized the mechanism lacking much evidence to support it. However, these can be areas for future research and must be carefully considered by the readers.

We agree that there are many areas of research that remain unclear, and we have now highlighted more of the knowledge gaps in the conclusions. Throughout our review we have, where appropriate, used language that alludes to the possibility of a mechanism, such as ‘suggests’, or the phrase ‘it may be possible’, to ensure the reader fully appreciates what is surmised by the literature.

  1. The authors have justly discussed the various players involved in the progression of diabetes, however the depth was missing.

We thank the reviewer for this comment. The authors appreciate that not every cell type or player in diabetes was discussed in-depth, however our goal was to bring together and highlight the dynamic range of possible cellular interactions at the islet level and therefore have focused much of our discussion on human pancreatic studies. To help the reader, we have signposted other reviews (for example, refs 27-29, 21, 65-68, 87, 88) that discuss specific cell types in more depth, which have been recently published.

  1. The artwork to demonstrate the interaction of beta cells and immune cells was very informative and summarized the body of work effectively.

The authors thank the reviewer for their positive comment regarding the artwork, and although minor adjustments have now been made, we feel this has only improved the summary of our review.

  1. One key feature that was lacking was the role of natural killer cells which also has been shown widely as vital players in diabetes development. Another missing link is how innate lymphoid cells (ILCs) play a role in the pathogenesis of type 1 diabetes. The authors have not discussed or even acknowledged these limitations.

We thank the reviewer for pointing out our oversight, for which we apologise.  We have now included some key insights into our new summary table and have briefly discussed NK cells, lines 303-314. In hindsight, we should have acknowledged other cell types that may have been less well noted in the tissue whilst stating that these could not be discussed in-depth, and as such, we have also included ILCs in our table, which is signposted is lines 237-241.

Reviewer 2 Report

Comments and Suggestions for Authors

This is an interesting and timely review that appears comprehensive and well-organized.  I have just a few suggestions that could enhance the presentation:

  1.  The authors do not discuss or speculate on the role of innate immune NK cells.  There is a literature on this topic in T1D, and it would be important to discuss their potential role, if any.
  2. The Figures are a bit complex and may be trying to do too much. The complexity of cell types, denoted by a visual legend, is difficult to follow. Figure 1, in particular, feels overdone, and while I am sure the authors can easily follow the figures they made, it is not so easy for the reader. Simplicity here would be a virtue.
  3. The inclusion of a table dedicated to B cells seems just a bit odd in a review that otherwise highlights a multitude of cell types.  While certainly there is an interest in B cells since the study of Leete et al. (a study that is now nearly 9 years old), it is not clear to me that the advancement in the elucidation of B cells has been so significant as to warrant a dedicated table (in which only 9 references are cited).  A more helpful table to a reader might focus instead on the different cells types of insulitis that were discussed in the review and highlight some of their hypothesized roles at various phases of the disease in both mice and humans.
  4. Please correct the "error" statement on line 52 (an incomplete sentence)

Author Response

Responses are in blue text, under each numbered comment.

This is an interesting and timely review that appears comprehensive and well-organized.  I have just a few suggestions that could enhance the presentation:

  1.  The authors do not discuss or speculate on the role of innate immune NK cells.  There is a literature on this topic in T1D, and it would be important to discuss their potential role, if any.

We thank the reviewer for pointing out our oversight, for which we apologise. We have now included a portion of text which discusses NK cells and included these cells in our summary table.

  1. The Figures are a bit complex and may be trying to do too much. The complexity of cell types, denoted by a visual legend, is difficult to follow. Figure 1, in particular, feels overdone, and while I am sure the authors can easily follow the figures they made, it is not so easy for the reader. Simplicity here would be a virtue.

We thank the reviewer for their feedback on our figures. On reflection, we agree figure 1 was busy, and we have now simplified the artwork by removing some of the cells in the figure legend that were not required in the overall message, concentrated our efforts on the interactions between the cells. We believe that this is much improved.

  1. The inclusion of a table dedicated to B cells seems just a bit odd in a review that otherwise highlights a multitude of cell types.  While certainly there is an interest in B cells since the study of Leete et al. (a study that is now nearly 9 years old), it is not clear to me that the advancement in the elucidation of B cells has been so significant as to warrant a dedicated table (in which only 9 references are cited).  A more helpful table to a reader might focus instead on the different cells types of insulitis that were discussed in the review and highlight some of their hypothesized roles at various phases of the disease in both mice and humans.

We thank the reviewer for this comment. However, whilst considerable work has been untaken in the last 9 years globally, advancing our knowledge of the role of B cells in type 1 diabetes, it has been recently reviewed quite comprehensively, and therefore we wanted to only highlight some of the key milestones in this area. However, on reflection, we have altered the table to more accurately reflect the review as a whole and agree this is much improved. We have tabulated the different cell types and, as suggested, stated their implicated role in type 1 diabetes.

  1. Please correct the "error" statement on line 52 (an incomplete sentence)

The authors apologise for the error code that has appeared in the manuscript following upload to the online system. This has now been updated and now refers to Section 3 of the manuscript where details of resident macrophages are present.

Reviewer 3 Report

Comments and Suggestions for Authors

This review paper by Walker et al entitled “Tissue resident and infiltrating immune cells: their influence on the demise of beta cells in type 1 diabetes” explores the current understanding of cellular interaction between resident and infiltration immune cells and islet environment in the context of T1D. The topic selection is appropriate for understanding the knowledge in the field and a good fit for the special issue of “Immune Responses in Type 1 Diabetes”.  There are some minor concerns:

  1. This paper summarizes studies in both human and animal models. In some places it is not clear if the cited study was done with human or animal.  Need to clarify.
  2. Need to cite references when discussing all endocrine cells display elevated HLA, and discuss the possible reasons why only beta cells are targeted. The authors discussed viral infection being a reason for elevated HLA but evidence of viral infection was limited to beta cells. The reason for the discrepancy was not discussed.
  3. Figure 2, changes on beta cells are summarized only as the results of immune attack. The role of beta cells in T1D pathogenesis is not simply being the victim, as illustrated in Figure 1.  It will be helpful to add beta cell changes that evoke or affect the interaction with immune cells in the Pseudotime trajectory. 
  4. Although discussed throughout the paper, it will be helpful to have a summary paragraph addressing gaps in the knowledge in the concluding remarks section.
  5. Ref [7] was cited to discuss CD8 in exocrine tissue. That paper did not show exocrine CD8 data.
  6. A couple of non-peer reviewed papers were cited, one was cited repeatedly. Need to mention that these are not peer reviewed.
  7. Line 52 bold “ Error... comment: Was an unfinished version submitted?  

Author Response

Responses are in blue text, under each numbered comment.

This review paper by Walker et al entitled “Tissue resident and infiltrating immune cells: their influence on the demise of beta cells in type 1 diabetes” explores the current understanding of cellular interaction between resident and infiltration immune cells and islet environment in the context of T1D. The topic selection is appropriate for understanding the knowledge in the field and a good fit for the special issue of “Immune Responses in Type 1 Diabetes”.  There are some minor concerns:

  1. This paper summarizes studies in both human and animal models. In some places it is not clear if the cited study was done with human or animal.  Need to clarify.

The authors thank the reviewer for this comment, where this was not clear previously, we have now detailed the use of human or animal models throughout the manuscript. See lines 56, 66, 106, and 114, 355, and Table 1.

2. Need to cite references when discussing all endocrine cells display elevated HLA, and discuss the possible reasons why only beta cells are targeted. The authors discussed viral infection being a reason for elevated HLA but evidence of viral infection was limited to beta cells. The reason for the discrepancy was not discussed.

The authors thank the reviewer for this comment. We have acted upon this suggestion and expanded this paragraph. Line 131-139 includes further HLA discussion:

‘Interestingly, in islets with residual insulin, all endocrine cells display this elevated expression, and it is presumed this is driven by the upregulation of interferon-related signalling pathways, which may drive elevated HLA in either a paracrine or autocrine manner, and yet only the beta cells are targeted. It has been postulated that other endocrine cells are more resilient to immunological killing by virtue of having different anti-apoptotic machinery – for example, anti-apoptotic BCL2L1, is more abundant in alpha cells, versus higher levels of pro-apoptotic CHOP being expressed in beta cells [32], similar differences may exist across the other islet cell types.’

  1. Figure 2, changes on beta cells are summarized only as the results of immune attack. The role of beta cells in T1D pathogenesis is not simply being the victim, as illustrated in Figure 1.  It will be helpful to add beta cell changes that evoke or affect the interaction with immune cells in the Pseudotime trajectory. 

We thank the reviewer for this suggestion. We have now included changes in the beta cell to the pseudotime trajectory, and added a description of a study on TET2 in our text (lines 145-149) that we feel is appropriate to have noted in Figure 2.

4. Although discussed throughout the paper, it will be helpful to have a summary paragraph addressing gaps in the knowledge in the concluding remarks section.

We have now included some further discussion in our concluding remarks section. We have addressed some gaps in our current knowledge, specifically at the islet level.

5. Ref [7] was cited to discuss CD8 in exocrine tissue. That paper did not show exocrine CD8 data.

The authors apologise for this oversight and have removed this reference from this statement in line 45. The other references for this sentence have been checked and are relevant.

6. A couple of non-peer reviewed papers were cited, one was cited repeatedly. Need to mention that these are not peer reviewed.

The authors thank the reviewer for raising this to our attention. In the text we have noted the status of both reports, stating both are preprint articles. Huber et al. 2024 bioRXiv, we have noted is a preprint (Line 402), and Barlow et al. 2024 eLife we have noted is a peer-reviewed preprint with the peer-review comments available online (Line 291-292).

7. Line 52 bold “ Error...” comment: Was an unfinished version submitted?  

The authors apologise for the error code that has appeared in the manuscript following upload to the online system. This has now been updated and now refers to Section 3 of the manuscript where details of resident macrophages are present.

Round 2

Reviewer 1 Report

Comments and Suggestions for Authors

The authors have acknowledged the comments and responded to them. The responses, albeit concise, summarize the roles of the immune cell players involved in TD pathophysiology. Congratulations to the author for a well-drafted manuscript.